# *BrWAX3*, Encoding a β-ketoacyl-CoA Synthase, Plays an Essential Role in Cuticular Wax Biosynthesis in Chinese Cabbage

**DOI:** 10.3390/ijms231810938

**Published:** 2022-09-19

**Authors:** Shuangjuan Yang, Hao Tang, Xiaochun Wei, Yanyan Zhao, Zhiyong Wang, Henan Su, Liujing Niu, Yuxiang Yuan, Xiaowei Zhang

**Affiliations:** 1Institute of Horticulture, Henan Academy of Agricultural Sciences, Zhengzhou 450002, China; 2College of Horticulture and Landscape, Henan Institute of Science and Technology, Xinxiang 453003, China

**Keywords:** *Brassica rapa*, β-ketoacyl-CoA synthase, wax synthesis, map-based cloning

## Abstract

In this study, we identified a novel glossy mutant from Chinese cabbage, named SD369, and all wax monomers longer than 26 carbons were significantly decreased. Inheritance analysis revealed that the glossy trait of SD369 was controlled by a single recessive locus, *BrWAX3*. We fine-mapped the *BrWAX3* locus to an interval of 161.82 kb on chromosome A09. According to the annotated genome of *Brassica rapa*, *Bra024749* (*BrCER60.A09*), encoding a β-ketoacyl-CoA synthase, was identified as the candidate gene. Expression analysis showed that *BrCER60.A09* was significantly downregulated in all aerial organs of glossy plants. Subcellular localization indicated that the BrCER60.A09 protein functions in the endoplasmic reticulum. A 5567-bp insertion was identified in exon 1 of *BrCER60.A09* in SD369, which lead to a premature stop codon, thus causing a loss of function of the BrCER60.A09 enzyme. Moreover, comparative transcriptome analysis revealed that the ‘cutin, suberine, and wax biosynthesis’ pathway was significantly enriched, and genes involved in this pathway were almost upregulated in glossy plants. Further, two functional markers, BrWAX3-InDel and BrWAX3-KASP1, were developed and validated. Overall, these results provide a new information for the cuticular wax biosynthesis and provide applicable markers for marker-assisted selection (MAS)-based breeding of *Brassica rapa*.

## 1. Introduction

The lipidic cuticle exists on the aerial surface of many land plants, working as a physical barrier to prevent nonstomatal water loss [1,2]. The main components of cuticles are cutin and cuticular wax. Cutin is a cross-linked polymer of modified long-chain fatty acids (C16 and C18 hydroxy and epoxy fatty acids) and glycerol, which provides mechanical strength to the surface layer [3,4]. Cuticular wax is a mixture of very-long-chain fatty acids (VLCFAs) and their derivatives [5,6]. Cuticles also protect plants from various biotic and abiotic stresses [7,8], profoundly affect plant-insect interactions [9], affect the pollen-stigma signaling [10], and prevent epidermal fusions [11].

Wax biosynthesis begins with de novo C16 and C18 fatty acid biosynthesis in the plastids of epidermal cells, and further elongates into VLCFAs in the endoplasmic reticulum (ER). The VLCFAs are then modified via two pathways: the alkane-forming pathway and the alcohol-forming pathway. The former generates aldehydes, alkanes, secondary alcohols, and ketones, while the latter produces primary alcohols and wax esters [2,5,6]. As the essential precursors of wax, VLCFAs are synthesized under the consecutive catalyzes of four different enzymes, namely, β-ketoacyl-CoA synthase (KCS), β-ketoacyl-CoA reductase (KCR), β-hydroxyacyl-CoA dehydratase (PAS2/HCD), and enoyl-CoA reductase (ECR/CER10), which formed a fatty acid elongase (FAE) complex [5,6,12]. Among the enzymes, KCSs determine substrate specificity, and are key rate-limiting condensing enzymes. In *Arabidopsis*, a large family of 21 *KCS* genes has been annotated [13]. To date, several *KCSs* have been reported. *FAE1/KCS18* encodes a seed-specific condensing enzyme that catalyzes the elongation of C18 to C20 and C22 [14]. *KCS2* and *KCS20* are responsible for the elongation of C20 to C22 and are functionally redundant [15]. *KCS9* is involved in the elongation of C22 to C24 fatty acids, which play multiple roles in the production of suberins, cuticular waxes, and membrane lipids [16]. *KCS1* is required for the elongation of C24 VLCFAs [17]. *KCS5/CER60* and *KCS6/CER6* play redundant roles in the elongation of C26 to C28 during wax biosynthesis, among which *KCS6* (*CER6*) plays a major role [10,18,19,20]. Together with *KCS5/CER60* or *KCS6/CER6*, three BAHD acyltransferases members, CER2, CER2-like1, and CER2-like2, participate in the synthesis of VLCFAs longer than C28 [9,21,22].

For economically important leafy vegetables in *Brassica* species, such as *Brassica rapa* (A genome) and *Brassica oleracea* (C genome), the waxless and glossy phenotype is an important commodity trait in breeding [23,24]. Therefore, identifying genes responsible for the glossy phenotype facilitates the breeding of new varieties with a glossy green phenotype. In *Brassica oleracea*, *Bol018504* (*CER1*) is responsible for the glossy trait of four materials, in which the alkane-forming pathway of wax biosynthesis is significantly affected [4,25,26]. *Bol013612* (*CER4*) is the candidate gene in two glossy materials, in which the primary alcohols and wax esters from the alcohol-forming pathway are severely reduced [27,28]. Recently, Ji et al. (2021) identified a new glossy cabbage genotype and found that *Bo1g039030* (*BoCER2*) was the candidate gene, in which wax monomers longer than C28 were significantly decreased [23]. In *Brassica rapa*, only two wax biosynthesis-related genes have been identified. *BrWax1* was first reported and *Bra013809* (*CER2*) was the candidate gene [29]. *Bra032670* (*CER1*), with different sequence variations, was responsible for the glossy trait in three *B.rapa* materials [24,30,31]. 

In this study, we characterized a novel glossy green Chinese cabbage mutant, SD369, which showed a significant reduction of wax monomers longer than C26. Genetic analysis suggested that the glossy trait was controlled by a single recessive gene. Map-based cloning revealed that the *Bra024749* gene, which is homologous to *CER60* (*KCS5*) in *Arabidopsis*, was the candidate gene, which has never been reported in *Brassica* species. Furthermore, sequence analysis and expression analysis showed that a 5567-bp insertion blocked VLCFA elongation from C26 to C28 in SD369, thus causing a wax deficiency. Additionally, we developed and validated two functional markers. These findings will provide new insight into the plant cuticular wax metabolic networks and will promote molecular marker-assisted breeding in *B.rapa*.

## 2. Results

### 2.1. Phenotypic Characterization and Genetic Analysis of Glossy Trait in SD369

SD369 is a spontaneous wax-deficient mutant found in the Chinese cabbage field. In contrast to the typical waxy appearance of R16-11 (P_2_), all aerial organs of SD369 (P_1_), such as the leaves (Figure 1a), stems (Figure 1b), flower buds (Figure 1c), and seedpods, were glossy green. Cryo-SEM analysis revealed many more wax crystals on R16-11 (Figure 1h) than on the SD369 (Figure 1d). The wax crystals on R16-11 were mainly flaky and columnar. However, the leaves of SD369 were covered with only a few wax crystals (Figure 1d), and the wax crystal shape was granular. 

F_1_, F_2_, BC_1_P_1_, and BC_1_P_2_ populations were generated to investigate the SD369 glossy trait inheritance. The F_1_ plants were all waxy, indicating that the waxy phenotype was dominant in the glossy phenotype. In a small F_2_ population, 102 plants were waxy and 40 were glossy, corresponding to a segregation ratio of 3:1 by the chi-square test (Table 1). A larger F_2_ population showed similar results (3026 waxy: 954 glossy, χ^2^ = 2.25 < χ^2^_0.05_ = 3.84). A ratio of 1:1 (540 waxy:494 glossy, χ^2^ = 2.05 < χ^2^_0.05_ = 3.84) was obtained in the BC_1_ P_1_ population, while in the BC_1_ P_2_ population, all 200 individuals were waxy (Table 1). These results indicated that the glossy phenotype of SD369 is controlled by a single recessive gene (Table 1). We tentatively named this locus *BrWAX3*.

### 2.2. Cuticular Wax Analysis via GC-MS

To investigate the reason of wax depletion in SD369, cuticular wax from W-bulk and G-bulk was examined by GC-MS. The wax load on waxy leaves reached, on average, 149.67 µg per g fresh weight, whereas wax loads on glossy leaves were severely reduced, with an average of 48.56 µg/g fresh weight, which reduced 68% of the wax load when compared to W-bulk (Figure 2b, Appendix A). Wax composition analysis revealed that most products from alkane-forming pathways decreased severely in G-bulk. For instance, the C29 alkane, C30 aldehyde, and C29 ketones in G-bulk reached only 2.5, 0, and 4.4%, respectively, of the levels found on the leaves of W-bulk (Figure 2a, Appendix A). However, the C25 alkane and C26 aldehyde were significantly increased in G-bulk than in W-bulk (Figure 2a, Appendix A). Considering the products from the alcohol-forming pathway, the amount of C28 primary alcohol decreased by 62% in G-bulk, whereas the amount of C26 primary alcohol increased approximately 2.8-fold in G-bulk. Similarly, the amounts of C28 and C30 fatty acids decreased significantly, whereas the amount of C26 fatty acids increased significantly in G-bulk (Figure 2a, Appendix A). Overall, wax components with chain lengths beyond C26 decreased severely in glossy plants, while shorter chains increased several-fold compared with those in the waxy plants. These findings suggested that the glossy phenotype of SD369 might be caused by the interruption in VLCFA carbon-chain elongation from C26 to C28 during cuticular wax biosynthesis.

### 2.3. Fine Mapping of the BrWAX3 Gene

To identify candidates of the *BrWAX3* gene, 50 waxy (W-pool) and 50 glossy (G-pool) individuals were selected from the F_2_ population and used to construct two extreme pools for Bulked Segregant Analysis (BSA). In total, we obtained 168 and 173 million raw reads for the W-pool and G-pool (Appendix A), and 1418,060 SNPs and 201,519 InDels were identified between the two DNA pools. Through sliding window analysis with the absolute value of Δ(SNP-index), a 6.5-Mb candidate region from 19.65 to 26.15 Mb on chromosome A09 was identified at a 0.01 confidence level (Figure 3a). 

Based on BSA-seq analysis results, 56 KASP markers were developed, and 20 markers (Appendix A) were polymorphic between the parents. Using these 20 KASP markers, 93 F_2_ plants were genotyped for linkage analysis (Appendix A). As shown in Figure 3b, the *BrWAX3* locus was initially mapped to a region on chromosome A09 between KASP markers LF3-K47 and LF3-K56, with a physical interval of 232 kb. The genetic distances between the *BrWAX3* locus and LF3-K47 and LF3-K56 were 0.1 and 0.5 cM, respectively (Figure 3b). 

To further finely map the *BrWAX3* locus, 954 glossy F_2_ plants were screened using flanking markers, LF3-K33 and LF3-K51, and a total of 115 recombinants (type 1 and type 10) were identified (Figure 3c). All the 115 recombinants were further genotyped using LF3-K36, LF3-K37, LF3-K40, LF3-K42, LF3-K46, LF3-K47, LF3-K56, and LF3-4K9, based on which 6 recombinants (type 6 and type 8) were identified (Figure 3c). Then, 4 KASP markers and 7 sequencing markers were further developed to genotype the 6 recombinants. The results delimited the *BrWAX3* gene to a 161.82-kb interval between markers LF3-seq8 and LF3-K60 (type 7 and type 8) (Figure 3c), with 1 recombinant between *BrWAX3* and LF3-seq8 and 3 recombinants between *BrWAX3* and LF3-K60. Five markers, namely, LF3-seq7, LF3-seq2, LF3-seq3, LF3-seq5, and LF3-seq9, co-segregated with the *BrWAX3* gene in the fine-mapping population (Figure 3c).

### 2.4. Candidate Gene Analysis

According to the annotation of *B.rapa* reference genome (V1.5), a total of 16 annotated or predicted genes were found within the 161.82-kb candidate interval (Table 2). Among these 16 genes, only the *Bra024749* gene, which is homologous to *CER60* in *Arabidopsis*, could be the candidate gene (Table 2). *CER60* encodes a β-ketoacyl-CoA synthase that is involved in the biosynthesis of very long-chain fatty acids (VLCFAs) during cuticular wax biosynthesis. 

The genomic sequence (gDNA) and coding sequence (CDS) of *Bra024749* in the parental lines were amplified and sequenced using the primer pairs BrWAX3-Ful-F and BrWAX3-Ful-R1 (Appendix A). The results showed that the *Bra024749* gene in the waxy parent R16-11 was 2112 bp in length and contained 2 exons and 1 intron (Figure 4a). The CDS of the *Bra024749* gene in R16-11 was 1494 bp in length, which shares 87% identity with *CER60* (*KCS5*) in *Arabidopsis*. Therefore, *Bra024749* was also designated *BrCER60.A09* in this study. However, in the glossy parent SD369, the *Bra024749* gene was 7679 bp in length, which was caused by a 5567-bp insertion at 590 bp in the first exon (Figure 4a,c). The large fragment insertion disrupted the normal transcription and translation of *Bra024749* in SD369. As shown in Figure 4a, no CDS products were detected in glossy SD369 using full-length primer pairs BrWAX3-Ful-F and BrWAX3-Ful-R1. Five primer pairs (P2-P6) (Figure 4b and Appendix A) spanning the full length of *Bra024749* in SD369 were further designed. Two of the five cDNA products, amplified using primers BrWAX3-P3 and BrWAX3-P5 (Figure 4b), could not be detected, which explained why the full-length CDS of *Bra024749* in SD369 could not be detected when the full-length primer pairs were used. We also mapped the RNA sequencing (RNA-seq) reads from G-bulk to the *Bra024749* gDNA sequence of SD369, and the results showed that very few reads were mapped to the 3200–3450 bp region and the 5500–5650 bp region (Appendix A), among the amplified regions of primers BrWAX3-P3 and BrWAX3-P5, respectively, which supported that transcription of *Bra024749* in SD369 was interrupted. Most importantly, the 5567-bp insertion caused premature translation termination at the 205 amino acid position, which caused the loss of function of BrWAX3 (Figure 4c,d). 

Based on the 5567-bp insertion, a functional marker BrWAX3-InDel (primer pairs BrWAX3-InDel-F and BrWAX3-InDel-R, Appendix A), which could amplify a 222-bp and 5789-bp product from lines R16-11 and SD369, respectively, was developed. When Phanta^®^ High-Fidelity DNA Polymerase (5 s/kb amplification rate) (Vazyme, Nanjing, China) with 30 s PCR extension time was used, all glossy F_2_ individuals showed 5789-bp products, and waxy F_2_ individuals presented either a homozygous 222-bp band or both bands (Figure 4e), which revealed that BrWAX3-InDel co-segregated with the cuticular wax phenotype in the F_2_ population. We also assayed the BrWAX3-InDel marker in the BC_1_P_1_ population and another F_2_ population (SD369 × SD2135)-F_2_ via EasyTaq DNA Polymerase (1 min/kb amplification rate) (Trans, Beijing, China) in conjunction with a 30 s PCR extension time. The results also showed 100% consistency between the cuticular wax phenotype and genotype (Appendix A) with no band in glossy individuals and a 222-bp band in waxy individuals, as the 30 s PCR extension time (1 min/kb amplification rate) was not enough for the 5789-bp product in glossy plants but was sufficient for the 222-bp product in waxy plants. We also developed and validated a KASP marker BrWAX3-KASP1 based on the 5567-bp insertion (Figure 4f,g), which could be used for high-throughput genotyping systems. 

Taken together, the above findings suggest that the *Bra024749* gene is the candidate gene for the cuticular wax gene *BrWAX3*.

### 2.5. Expression Pattern Analysis and Subcellular Localization of BrWAX3

The expression levels of *Bra024749* (*BrCER60.A09*) in different tissues of the two parent lines were examined by qRT-PCR analysis using primer pairs BrWAX3-qF and BrWAX3-qR (Appendix A). The results suggested that the *Bra024749* transcript was found in various tissues, including the stems, leaves, sepals, petals, stamens, and pistils, but not in the roots, with the highest level in leaves (Figure 5a). The expression of *Bra024749* was much lower in SD369 than in the waxy parent R16-11 in any of the tissues we examined (Figure 5a). 

To evaluate the subcellular localization, a fusion protein of BrCER60-GFP under the drive of 35 S CaMV promoter was transiently expressed in tobacco leaf epidermal cells. The results showed that the green fluorescent signals from BrCER60-GFP were found in the ER (Figure 6a), exactly overlapping with the red fluorescent signals from the ER marker (Figure 6b–d), indicating that *Bra024749* (*BrCER60.A09*) was localized to the ER. 

### 2.6. Sequence and Expression Pattern Analysis of BrCER6.A07

In *Arabidopsis*, *CER6* (*AT1G68530*) is a paralog of *CER60* (*AT1G25450*) [18]. Therefore, we blasted the coding sequence of *CER6* (*AT1G68530*) against the *B.rapa* genome, and found that the best-hit gene was *Bra004034* (*BrCER6.A07*), which shared 82.2% identity with the candidate *Bra024749* (*BrCER60.A09*) at the coding sequence level. We designed a gene-specific primer pair, BrCER6.A07-ful-F and BrCER6.A07-ful-R (Appendix A), to amplify the full-length CDS of *BrCER6.A07* in SD369 and R16-11. The CDSs of *BrCER6.A07* from SD369 and R16-11 were submitted to GeneBank under accession numbers OPO46432 and OPO46431. Eight SNPs were identified between the coding sequences of R16-11 and SD369 (Appendix A). Even though two SNPs caused nonsynonymous mutations, they did not affect protein function (Appendix A). We also compared the expression level of *BrCER6.A07* in the parental lines using primer pairs BrCER6.A07-qF and BrCER6.A07-qR (Appendix A). As shown in Figure 5b, the stamen showed the highest expression levels, whereas much lower levels were found in stems and leaves. Furthermore, *BrCER6.A07* showed a comparable level in stems between SD369 and R16-11 and was lower in leaves of waxy R16-11, which suggested that the *BrCER6.A07* gene was not responsible for glossy phenotype.

### 2.7. Transcriptome Analysis in Waxy and Glossy Stems

We performed comparative transcriptome analysis between the W-bulk and G-bulk to identify the gene regulatory networks involved in cuticular wax biosynthesis. We obtained approximately 261 million raw reads for the six cDNA libraries, ranging from 42.3 to 44.4 million reads per library (Appendix A). The raw data were submitted to SRA database under accession number PRJNA860219 (accessed on 19 July 2022). Among the clean reads, 75.1–79.7% were uniquely mapped to the reference genome (Appendix A). In total, we identified 5314 differentially expressed genes (DEGs) between the W-bulk and G-bulk, among which 2513 genes were upregulated and 2801 were downregulated in the G-bulk compared with the W-bulk. 

KEGG pathway enrichment analysis revealed that ‘cutin, suberine, and wax biosynthesis’ was significantly enriched (Figure 7a). In accordance with the reduced amount of cuticular wax in glossy plants, the candidate gene *Bra024749* (*BrCER60.A09*) was significantly downregulated in G-bulk, but most of the other genes involved in wax biosynthesis and its regulation were upregulated (Figure 7b, Appendix A). For example, the genes *LACS2* (*Bra032284*), *ECR* (*Bra007154*), *CER2* (*Bra013809*), *KCS2* (*Bra015296*), and *KCS20* (*Bra033694*), which participate in fatty acid elongation, were upregulated in G-bulk (Figure 7b, Appendix A). Most of the genes in alkane-forming pathways, such as *CER3* (*Bra002692*), *MAH1* (*Bra027907, Bra027906, Bra027904, Bra027898* and *Bra027897*), and genes in alcohol-forming pathway, such as *CER4* (*Bra011470*) and *WSD1* (*Bra000019*), were all upregulated in glossy plants (Figure 7b, Appendix A). Additionally, the *SHINE1* (*Bra026140*), *SHINE3* (*Bra009837* and *Bra036543*), *MYB30* (*Bra033067*, *Bra039040* and *Bra025361*) and *MYB106* (*Bra039140*) genes, which participate in transcriptional regulation of cutin and wax biosynthesis, were also upregulated in G-bulk (Figure 7b, Appendix A). Our qRT-PCR analysis of wax metabolism-related genes was consistent with the RNA-seq results (Appendix A). Cuticular waxes contain not only VLCFAs and their derivatives, but also other secondary metabolites, such as terpenoids, sterols, and flavonoids [6,24,32]. As expected, pathways of ‘flavonoid biosynthesis’, ‘phenylalanine metabolism’, ‘glucosinolate biosynthesis’, ‘stilbenoid, diarylheptanoid and gingerol biosynthesis’ were significantly enriched (Figure 7a), and most genes in these pathways were downregulated in G-bulk (Appendix A), which was consistent with the reduced number of waxes in glossy plants.

## 3. Discussion

In the present study, the *BrWAX3* gene, which confers wax biosynthesis, was successfully and finely mapped to a physical interval of 161.82 kb. Several lines of evidence indicate that *Bra024749* (*BrCER60.A09*) is the candidate gene for *BrWAX3*. (1) Among the 16 annotated genes within the 161.82 kb interval, only one gene, *Bra024749*, which is homologous to *CER60/KCS5* in *Arabidopsis*, might be involved in cuticular wax biosynthesis. (2) The expression level of *BrCER60.A09* in all aerial organs was much lower in glossy plants than in waxy plants. (3) A 5567-bp insertion was found in glossy plants, which resulted in a premature stop codon and loss of function of the CER60 enzyme. (4) Two functional markers for *BrCER60.A09* co-segregated with the wax phenotype. (5) Subcellular localization analysis showed that the *BrCER60.A09* was localized to the ER, which is the site of wax biosynthesis. (6) Cuticular wax composition analysis showed a reduction of wax monomers with chain lengths beyond C26 and an increased proportion of shorter chains, which was in agreement with the CER60 function in *Arabidopsis* [19]. Overall, the 5567-bp insertion of *BrCER60.A09* in SD369 was the main cause of the glossy phenotype. 

Studies in *Arabidopsis* showed that several *cer* mutants (*cer1, cer2, cer3,* and *cer6*) are male sterile due to defective pollen recognition or failed pollen hydration [18,33,34]. *CER6/KCS6* is involved in VLCFA elongation of C26 to C28 during cuticular wax biosynthesis. The *cer6* mutant showed a substantial reduction of derivatives beyond C26, nearly abolished stem wax accumulation, and exhibited conditional male sterility [10,18,19,20]. *CER60/KCS5*, a paralog of *CER6/KCS6*, plays a redundant role with *CER6/KCS6* in wax biosynthesis, but *CER6/KCS6* plays a major role [19]. The mutation in *CER60/KCS5* caused only a slight reduction in total wax amounts in leaves and flowers, and the wax amounts in stems barely changed [19]. In our study, the mutation of *BrWAX3* (*BrCER60.A09*) in SD369 caused a significant reduction of wax monomers with chain lengths beyond C26 and an increased number of shorter chains, which was the same as *CER60/KCS5* in *Arabidopsis* [19]. However, *BrWAX3* (*BrCER60.A09*) showed a higher expression level in stems and leaves than *Bra004034* (*BrCER6.A07*) did (Figure 5a and Appendix A), and the mutation of *BrWAX3* (*BrCER60.A09*) in SD369 caused an obvious glossy appearance of its stems, leaves, and flower buds, reflecting the predominant role of *BrWAX3* (*BrCER60.A09*) in wax biosynthesis on aerial organs in Chinese cabbage. These results were different from those for *CER60/KCS5* in *Arabidopsis*, which is transcribed at a low level in all the mature shoot tissues [20] and plays a minor role in wax biosynthesis [19]. However, another study revealed that GUS expression driven by the *AtKCS5* promoter was much higher than that driven by *AtKCS6* [35], which is consistent with the results of our study. Additionally, we obtained many seeds after self-pollination of SD369, suggesting that the mutation of *BrWAX3* (*BrCER60.A09*) in glossy SD369 did not cause male sterility, which was different from the male sterility that occurred for the *cer6* mutant [10,18]. We speculated that the higher expression of *BrCER6.A07* in stamens might compensate for the loss of function of *BrWAX3* (*BrCER60.A09*) in stamens of SD369 and restore the fertility. Further studies, such as those involving the generation of *BrWAX3* (*BrCER60.A09*) and *BrCER6.A07* single and double mutants, the identification of their possible roles in wax biosynthesis, and functional analysis of these two genes, are needed to provide more evidence for the involvement of these genes in wax formation and pollen development in *B.rapa*. 

The SD369 mutant and the cloned *BrWAX3* gene are important for breeding *B.rapa* varieties. *B.rapa* includes a variety of vegetables, such as Chinese cabbage, Pakchoi, Cai-xin, and purple cai-tai. The edible parts of Cai-xin and purple cai-tai are tender stems, for which the glossy appearance is preferred by customers [30]. Therefore, breeders can introduce the locus from SD369 to Cai-xin and purple cai-tai, which would create new varieties with glossy phenotype but does not influence male fertility (unlike the *cer1* mutant in *B.rapa*). We also developed two functional markers, BrWAX3-InDel and BrWAX3-KASP1, for *BrWAX3* (*BrCER60.A09*) based on the 5567-bp insertion, which could be used for molecular marker-assisted breeding either through agarose gel electrophoresis or by high-throughput genotyping platforms. 

The global effect of *BrWAX3* (*BrCER60.A09*) mutation on gene expression in Chinese cabbage was also investigated via comparative transcriptome analysis. In contrast to the lower expression level of *BrWAX3* (*BrCER60.A09*) in glossy plants, most genes in the fatty acid elongation, alkane-forming, and alcohol-forming pathways of wax biosynthesis, and in the transcriptional and posttranscriptional regulation of wax biosynthesis, were upregulated in glossy plants, indicating that a feedback mechanism occurred in glossy plants, which was consistent with the feedback observed in *nwgl* glossy cabbage [25]. However, we could not confirm it was the downregulation of *BrWAX3* (*BrCER60.A09*) or the reduced amount of wax that caused the feedback. Additionally, genes involved in cutin biosynthesis, such as *CYP77A6* (*Bra029852*) and *CYP86A4* (*Bra032642*), were also upregulated in glossy plants (Appendix A), which are similar to those in our previous studies [24].

## 4. Materials and Methods

### 4.1. Plant Materials

SD369 (P_1_), a doubled haploid (DH) line of Chinese cabbage with glossy phenotype, and R16-11 (P_2_), a DH line with wax phenotype, were used as the parents to construct the F_1_, F_2_, and BC_1_ populations for inheritance analysis and map-based cloning. Additionally, another F_2_ population, (SD369 × SD2135)-F_2_, was generated for marker validation by crossing the glossy line SD369 with the waxy line SD2135. All generations were grown in open fields at Henan Academy of Agricultural Sciences. At the bolting stage, the glossy and waxy phenotypes were assessed visually. Chi-square test (*χ*^2^) was used to examine the phenotype segregation ratios. 

### 4.2. Cryo-Scanning Electron Microscopy (cryo-SEM) and Gas Chromatography-Mass Mpectrometry (GC-MS)

A Hitachi SU3500 (Japan) scanning electron microscope was used to observe the abundance and morphology of wax of leaves from SD369 and R16-11. The method for cryo-SEM was followed as in our previous study [24].

W-bulk and G-bulk were constructed by mixing equal amounts of leaves from 10 waxy or 10 glossy F_2_ plants, respectively. In total, three biological replicates of W-bulk and G-bulk were constructed. The cuticular wax composition and components in W-bulk and G-bulk were assessed via GC-MS at Shanghai Jiao Tong University, which was performed as described in our previous study [24].

### 4.3. Identification of Candidate Genes via Bulked-Segregant Analysis Sequencing (BSA-Seq) and Kompetitive Allele-Specific PCR (KASP) Assays

Candidate genes were identified according to the BSA-seq method [36]. Two DNA pools were constructed by mixing equal amounts of DNA from 50 waxy F_2_ individuals (W-pool) and 50 glossy F_2_ individuals (G-pool). The two DNA pools were resequenced by Anoroad Biotech Co., Ltd. (Beijing, China) using 150-bp PE strategy. The raw data were deposited in the Sequence Read Archive (SRA) in NCBI as PRJNA859942. The Burrows-Wheeler Aligner (BWA) [37] was used to map the clean reads to the *B.rapa* reference genome V1.5 [38]. The SAMtools software (V1.3.1) [39] was used to detect the single-nucleotide polymorphism (SNP) and insertion/deletion (InDel) variants between the W-pool and G-pool. Then, we calculated the SNP-index and Δ(SNP-index) for all genomic positions in the W-pool and G-pool, which was performed as in our previous studies [24,40]. Finally, the absolute value of Δ(SNP-index) was calculated for sliding window analysis, with a 1-Mb window width and a 50-kb sliding window step.

We used KASP assay to conduct the initial linkage analysis of the *BrWAX3* gene, which was performed as our previous studies [24,40,41]. First, SNPs showing polymorphism between the two DNA pools and nearing the candidate BSA-seq region were selected for KASP marker development [40]. Then, KASP markers (Appendix A), showing polymorphism between the two parents, were employed to genotype the F_2_ population containing 93 individuals. The genetic linkage map was constructed using JoinMap 4.0 software [42], and followed as in our previous studies [24,40].

954 individuals with glossy phenotypes were used for fine mapping of the candidate gene.

### 4.4. Gene Cloning and Sequence Analysis

*BrCER60.A09*, the candidate gene of *BrWAX3*, and its homologue *BrCER6.A07*, were cloned using Phanta^®^ High-Fidelity enzyme Mix (Vazyme, Nanjing, China) in a total volume of 50 μL reaction: 3 μL DNA template, 3 μL of both forward and reverse primers, 25 μL enzyme mix, and 16 μL ddH_2_O. The PCR conditions were performed as in the manuals. The PCR products were sequenced by Sunya Biotech Co., Ltd. (Zhengzhou, China). The sequences of SD369 and R16 were aligned using DNAMAN.

### 4.5. RNA Extraction and Expression Analysis

Various tissue samples (root, stem, leaf, sepal, petal, stamen, and pistil) of SD369 and R16-11 were collected. The total RNA of each sample was extracted using RNAiso Plus reagent (TaKaRa, Japan), and the first-strand cDNA was synthesized using the TransScript One-Step gDNA Removal and cDNA Synthesis Kit (Trans, Beijing, China). Quantitative real-time PCR (qRT-PCR) was performed with SYBR Premix Ex TaqTM II (TaKaRa, Japan). The analysis of gene relative expression data was performed using the 2^-ΔΔCt^ method [43]. *BrGAPDH* was employed as the reference gene [24,40]. The primers are listed in Appendix A.

### 4.6. Subcellular Localization

The coding sequences of *BrCER60.A09* without the stop codon were amplified from waxy R16-11 using the primer pairs BrWAX3-fulF and BrWAX3-fulR2 (Appendix A). The amplified cDNA fragments were subsequently inserted into the modified pBWA(V)HS vector under the control of the 35 S promoter. The resultant binary plasmid was referred to as pBrCER60-GFP. A pAtPIN5-RFP construct was used as an ER marker [44]. Vector of pBrCER60-GFP, pAtPIN5-RFP, and the blank control vector were infiltrated into epidermal cells of tobacco leaves through Agrobacterium-mediated transformation. Fluorescence signals were observed with a confocal laser scanning microscope (Nikon C1, Japan).

### 4.7. Transcriptome Analysis

The W-bulk and G-bulk each with three replicates were subjected to mRNA sequencing by BioMarker Tech Co., Ltd. (Beijing, China). The clean reads of each sample were aligned to the *B.rapa* reference genome (V1.5) using HISAT2 software (V2.1.0) [45]. Then, the fragments per kilobase of transcript per million mapped reads (FPKM) value of each gene were calculated to estimate gene expression levels. Genes with a *q*-value ≤ 0.05 and |log2(fold change)| ≥ 1 identified by DESeq2 (V1.6.3) [46] were recognized as differentially expressed genes (DEGs). Kyoto Encyclopedia of Genes and Genomes (KEGG) pathway enrichment analysis was implemented using TBtools [47].

## 5. Conclusions

The present study showed the molecular mechanism of wax deficiency in SD369. The *BrWAX3* locus was fine-mapped to an interval of 161.82 kb, and *Bra024749* (*BrCER60.A09*), which encodes a β-ketoacyl-CoA synthase, was the most likely candidate gene for *BrWAX3*. A 5567-bp insertion of *BrWAX3* (*BrCER60.A09*) in glossy SD369 caused a loss of protein function, thus blocking the VLCFA elongation of C26 to C28, and ultimately resulting in the glossy phenotype. The loss of function of *BrWAX3* (*BrCER60.A09*) in glossy plants also caused feedback of genes involved in cutin and wax biosynthesis pathways. Besides, two functional markers for *BrWAX3* were developed and validated. Our research will promote molecular research on wax synthesis in *Brassica rapa*.

## Figures and Tables

**Figure 1 ijms-23-10938-f001:**
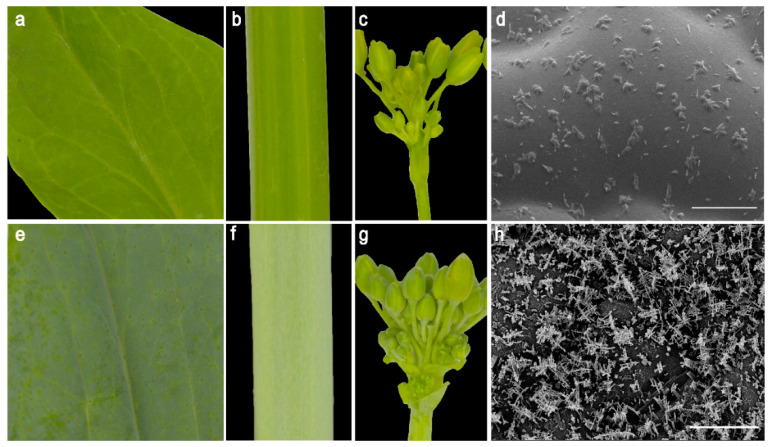
Phenotypic characterization of cuticular waxes in the two parent lines (SD369 and R16-11). The stem (**a**), leaf (**b**), and flower buds (**c**) of SD369 showed glossy phenotype at the bolting stage, as compared to the waxy appearance of R16-11 stem (**e**), leaf (**f**), and flower buds (**g**). Cryo-scanning electron microscopy images of leaves from SD369 (**d**) and R16-11 (**h**). Bar = 10 μm in (**d**,**h**).

**Figure 2 ijms-23-10938-f002:**
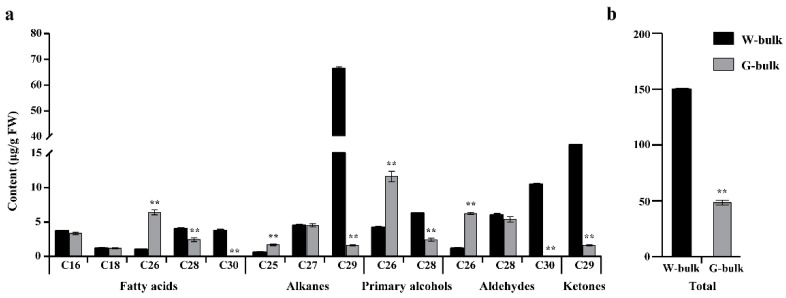
Cuticular wax composition in leaves from W-bulk and G-bulk. (**a**) Wax composition of W-bulk and G-bulk. (**b**) Total wax load of W-bulk and G-bulk. Error bars indicate SD (n = 3). ** *p*-value < 0.01.

**Figure 3 ijms-23-10938-f003:**
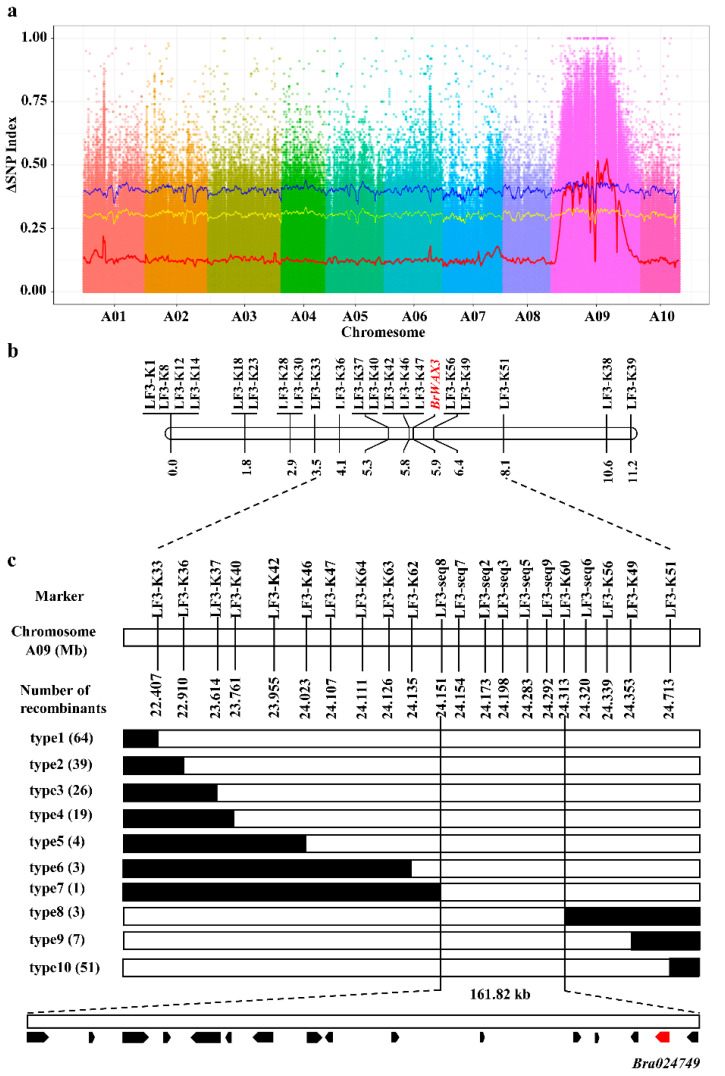
Map-based cloning of *BrWAX3* gene in Chinese cabbage. (**a**) BSA-seq analysis for *BrWAX3*. The Δ(SNP-index) was calculated at 1-Mb intervals with a 50-kb sliding window. One candidate region was identified on chromosome A09. (**b**) Initial mapping of *BrWAX3*. The genetic map of *BrWAX3* was showed with cM as the unit. (**c**) Fine mapping of *BrWAX3*. The *BrWAX3* gene was delimited to an interval between LF3-seq8 and LF3-K60 on chromosome A09, with an estimated physical length of 161.82 kb, and 16 genes were annotated in this region based on the reference genome sequence. The genetic structure of each recombinant type is depicted as white for homozygous glossy phenotype, black for heterozygous alleles, respectively. The number of each recombinant type is indicated in the brackets.

**Figure 4 ijms-23-10938-f004:**
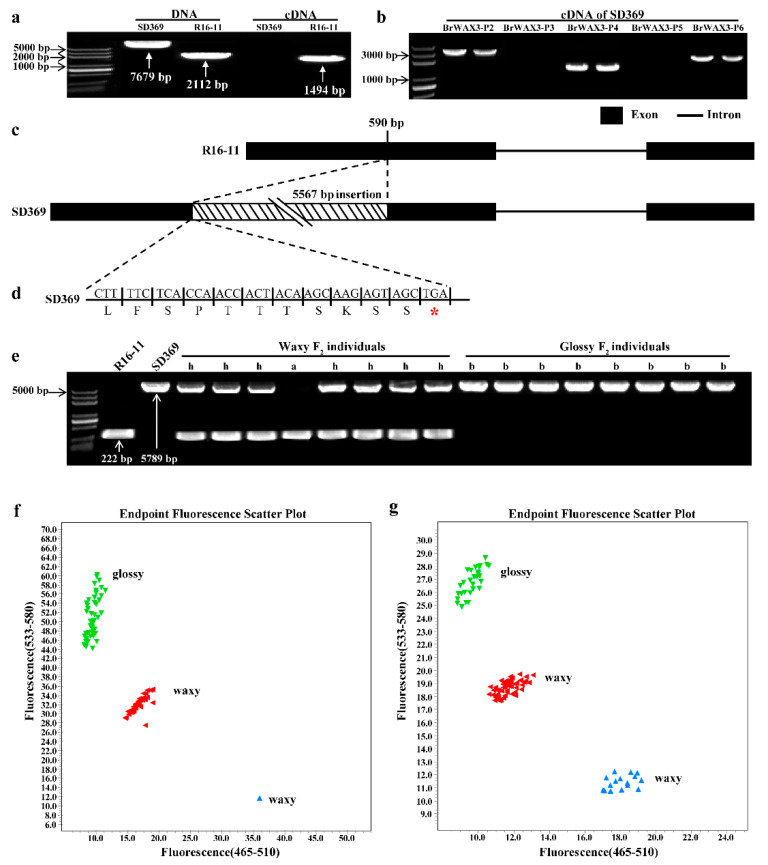
Candidate gene analysis of *BrWAX3*. (**a**) Amplification products of the full length primer using DNA and cDNA from SD369 and R16-11. The full length CDS of glossy SD369 could not be detected. (**b**) Amplification products of each fragmental primer using cDNA from SD369. (**c**) *BrWAX3* includes 2 exons and 1 intron. (**d**) A 5567-bp insertion in glossy SD369 caused a premature stop codon. (**e**) Validation of the functional marker BrWAX3-InDel in F_2_ individuals. (**f**,**g**) Validation of the functional marker BrWAX3-KASP1 in BC_1_P_1_ population (**f**) and in (SD369 × SD2135)-F_2_ population (**g**).

**Figure 5 ijms-23-10938-f005:**
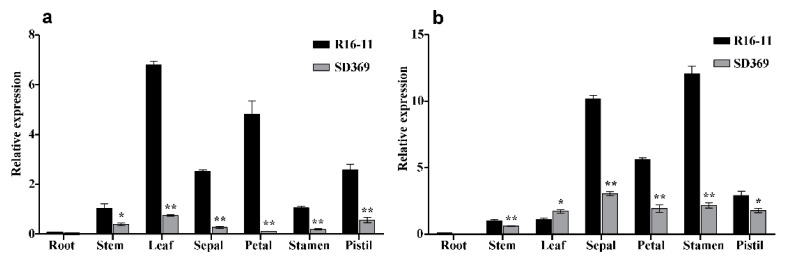
Gene expression data analysis. Quantitative RT-PCR of *BrWAX3* (*BrCER60.A09*) (**a**) and *Bra004034* (*BrCER6.A07*) (**b**) in different tissues of the two parents. The *BrGAPDH* was used as an internal control. Error bars indicate SE (n = 3). * *p* < 0.05. ** *p* < 0.01.

**Figure 6 ijms-23-10938-f006:**
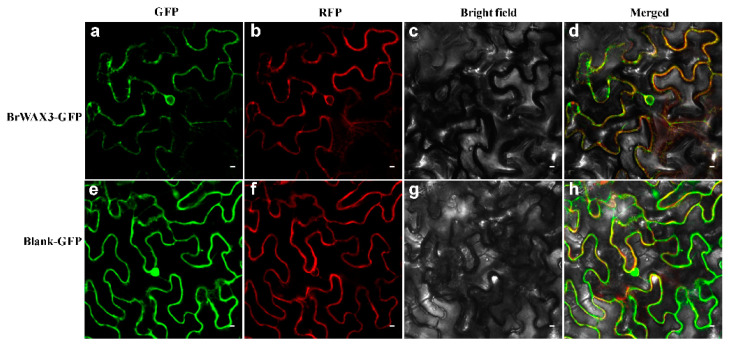
Subcellular localization analysis of BrWAX3 (BrCER60.A09). (**a**,**e**) The fusion construct BrWAX3-GFP (**a**) and the construct control (**e**) were transiently introduced into the tobacco leaf epidermal cells. (**b**,**f**) Fluorescence signals of an ER marker protein AtPIN5 fused with RFP. (**c**,**g**) Bright-field image. (**d**,**h**) Merged image of bright field and fluorescence.

**Figure 7 ijms-23-10938-f007:**
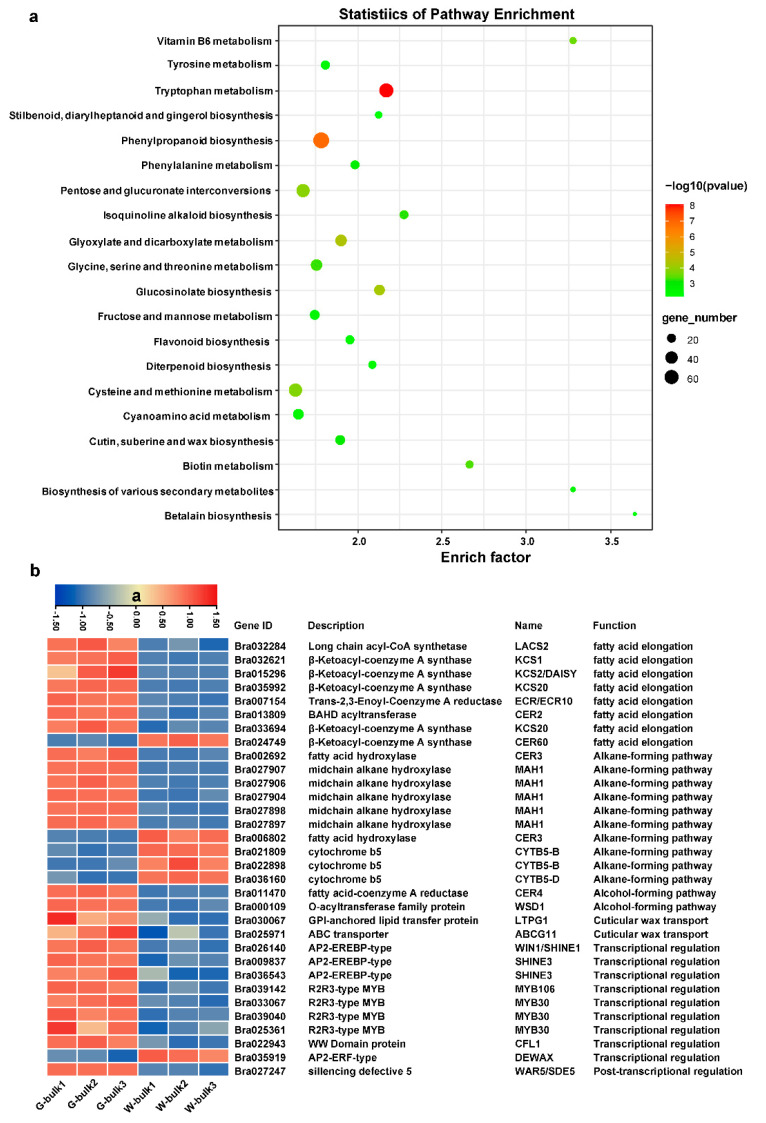
Transcriptome analysis in waxy and glossy stems. (**a**) Scatter plot of top 20 enriched KEGG pathways. Rich factor is the ratio of the DEG number to the background number in a certain pathway. The size of the dots represents the number of genes, and the color of the dots represents the range of the *p*-value. (**b**) Differentially expressed genes involved in cuticular wax metabolism. The heatmap colors are shown in log2(FPKM + 1). Three biological replicates of the W-bulk and G-bulk are shown.

**Table 1 ijms-23-10938-t001:** Genetic analysis of the glossy trait in crosses between SD369 and R16-11.

Population	Total	Waxy	Glossy	Expected ratio	χ^2^	χ^2^_0.0__5_
P_1_ (SD369)	10	10	0	-	-	-
P_2_ (R16-11)	10	0	10	-	-	-
F_1_	15	15	0	-	-	-
F_2_-small	142	102	40	3:1	0.76	3.84
F_2_-large	3980	3026	954	3:1	2.25	3.84
BC_1_P_1_ (F_1_ × SD369)	1020	540	494	1:1	2.05	3.84
BC_1_P_2_ (F_1_ × R16-11)	300	300	0	-	-	-

**Table 2 ijms-23-10938-t002:** Annotated genes in the candidate interval of the *BrWAX3* locus.

Gene Name	Gene Position on A09	*Arabidopsis* Homolog	Gene Function
*Bra024763*	24151883-24153849	*AT1G25280*	Member of TLP family
*Bra024762*	24164515-24164946	*AT5G05020*	Pollen Ole e 1 allergen and extension family protein
*Bra024761*	24171584-24174217	*AT1G25320*	Leucine-rich repeat protein kinase family protein
*Bra024760*	24179389-24180542	*AT1G25340*	putative transcription factor (MYB116)
*Bra024759*	24185650-24190615	*AT1G25350*	glutamine-tRNA ligase, putative/glutaminyl-tRNA synthetase
*Bra024758*	24192301-24193325	*AT1G25370*	hypothetical protein (DUF1639)
*Bra024757*	24197064-24199869	*AT1G25375*	Metallo-hydrolase/oxidoreductase superfamily protein
*Bra024756*	24207077-24209333	*AT1G25380*	Encodes a mitochondrial-localized NAD+ transporter that transports NAD+ in a counter exchange mode with ADP and AMP in vitro
*Bra024755*	24210118-24212198	*AT1G25390*	Protein kinase superfamily protein
*Bra024754*	24226168-24227349	*AT2G05970*	F-box protein (DUF295)
*Bra024753*	24253251-24253559	*AT1G25422*	hypothetical protein
*Bra024752*	24279278-24280968	*AT1G29470*	S-adenosyl-L-methionine-dependent methyltransferases superfamily protein
*Bra024751*	24285640-24285933	*AT1G25425*	CLAVATA3/ESR-RELATED 43
*Bra024750*	24299524-24301114	*AT1G25440*	B-box type zinc finger protein with CCT domain-containing protein
*Bra024749*	24304800-24306911	*AT1G25450*	Encodes KCS5, a member of the β-ketoacyl-CoA synthase family involved in the biosynthesis of VLCFA (very long chain fatty acids); CER60
*Bra024748*	24311235-24312823	*AT1G69710*	Regulator of chromosome condensation (RCC1) family with FYVE zinc finger domain-containing protein

## Data Availability

The raw data from BSA-seq analysis have been deposited into the SRA database (https://www.ncbi.nlm.nih.gov/sra/) with accession number PRJNA859942 (accessed on 18 July 2022). The raw data from transcriptome analysis were deposited in the SRA database under accession number PRJNA860219 (accessed on 19 July 2022). The CDS sequence of *BrWAX3* from waxy R16-11 was deposited in Genbank under accession number OPO46430 (accessed on 28 July 2022).

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
