# Peer review of "BrWAX3, Encoding a β-ketoacyl-CoA Synthase, Plays an Essential Role in Cuticular Wax Biosynthesis in Chinese Cabbage"

_ijms, 2022, doi:10.3390/ijms231810938_

Round 1
Reviewer 1 Report
In the current research article entitled " BrWAX3, Encoding a β-ketoacyl-CoA Synthase, Plays an Essential Role in Cuticular Wax Biosynthesis in Chinese Cabbage", Yang et al., evaluated a novel glossy mutant from Chinese cabbage and fine-mapped the BrWAX3 locus to an interval of 161.82 kb on chromosome A09, encoding a β-ketoacyl-CoA synthase. In this study authors found the expression of BrCER60.A09 was significantly downregulated in all aerial organs of glossy plants and localized in endoplasmic reticulum and a premature stop codon causing a loss of function of BrCER60.A09 enzyme. Moreover, comparative transcriptome analysis revealed that the 'cutin, suberine and wax biosynthesis' pathway was significantly enriched. Authors concluded that this new information could provide applicable markers for marker-assisted selection (MAS)-based breeding of Brassica rapa. This article addresses a research topic of great interest. However, this reviewer has certain suggestions that would help produce a more comprehensive overview of the topic:
Comments:
1. The English of manuscript can be polished (minor) and there are few typo errors in the manuscript that can be checked.
2, Did authors find any change in photosynthesis rate between SD369 and R16-1 lines?
3, Is there any difference to transpiration rate and pathogen resistance between SD369 and R16-1 lines?
4. The authors may additionally provide challenges, or prospect of the present study.
5. At least one illustrative figure may be provided as to highlight the summary of this study.
6. The authors should cross-check all abbreviations in the manuscript. Initially, define in full name followed by abbreviation.
Author Response
1,The English of manuscript can be polished (minor) and there are few typo errors in the manuscript that can be checked.
Answer:Thank you very much. Some of the typo errors have been revised as suggested.
2, Did authors find any change in photosynthesis rate between SD369 and R16-1 lines?
Answer:Thank you very much. It is a good idea to check if there was any change in photosynthesis rate between SD369 and R16-11 lines. Previous studies have showed that the chlorophyll (Chl) content and photosynthesis rate was lower in wax-less mutant compared to the WT (OsPLS4 Is Involved in Cuticular Wax Biosynthesis and Affects Leaf Senescence in Rice, 2020, Frontiers in Plant science). Considering the different background of SD369 and R16-11, We will analyze the chlorophyll (Chl) content and photosynthesis rate in the over-expression line of BrWAX3 and the SD369, or in the CRISPR/cas9 knockout of BrWAX3 and the R16-11, which is a major part of our following work.
3, Is there any difference to transpiration rate and pathogen resistance between SD369 and R16-1 lines?
Answer:Thank you very much. Previous studies have revealed that the transpiration rate was higher in wax-less mutant and the pathogen resistance was lower in wax-less mutant (Preliminary study of the characteristics of several glossy cabbage (Brassica oleracea var. capitata L) mutants, 2015, Horticutural Plant Journal; Advances in the understanding of cuticular waxes in Arabidopsis thaliana and crop species, 2015, Plant Cell Rep.), because cuticular wax plays an important role in protecting plants from various biotic and abiotic stresses. Considering the different background of SD369 and R16-11, We will analyze the transpiration rate and pathogen resistance in the over-expression line of BrWAX3 and the SD369, or in the CRISPR/cas9 knockout of BrWAX3 and the R16-11, which is also a major part of our following work.
4. The authors may additionally provide challenges, or prospect of the present study.
Answer:Thank you very much. We have already provide prospect of the present study in the end of second paragraph in "3. Discussion", which displayed as "Further studies, such as those involving the generation of BrWAX3 (BrCER60.A09) and BrCER6.A07 single and double mutants, the identification of their possible roles in wax biosynthesis, and functional analysis of these two genes, are needed to provide more evidence for the involvement of these genes in wax formation and pollen development in B.rapa."
5. At least one illustrative figure may be provided as to highlight the summary of this study.
Answer:Thank you very much. One illustrative figure was provided as suggested.
6,The authors should cross-check all abbreviations in the manuscript. Initially, define in full name followed by abbreviation.
Answer:Thank you very much. We have checked all abbreviations in the manuscripts as suggested.

Reviewer 2 Report
A study by Yang etal identified a novel glossy mutant from chinese cabbage. The study was comprehensive and addressed the reseacrh questions ver well. The results and methodologies are clearly described.
Minor comments:
1) The M and M part, section 4.5 and 4.7 can be combined in one section.
2) The quality of figure is relatively low.
Author Response
1) The M and M part, section 4.5 and 4.7 can be combined in one section.
Answer:Thank you very much. "Section 4.5" displayed the detailed methods about "RNA Extraction and Expression Analysis", "Section 4.7" displayed the detailed methods about "Transcriptome Analysis", which was accordance with the results part. So we would like to keep this two section. Thank you very much.
2) The quality of figure is relatively low.
Answer:Thank you for your comments. Actually, all the figures in our manuscript were 600 dpi, which met the figure requirements of "International Journal of Molecular Sciences". Thank you again.